# Burnout and Perceived Performance Among Junior Athletes—Associations with Affective and Cognitive Components of Stress

**DOI:** 10.3390/sports7070171

**Published:** 2019-07-11

**Authors:** Frode Moen, Maria Hrozanova, Tore C. Stiles, Frode Stenseng

**Affiliations:** 1Centre for Elite Sports Research, Department of Education and Lifelong Learning, Faculty of Social and Educational Sciences, Norwegian University of Science and Technology, 7491 Trondheim, Norway; 2Centre for Elite Sports Research, Department of Neuromedicine and Movement Science, Faculty of Medicine and Health Science, Norwegian University of Science and Technology, 7491 Trondheim, Norway; 3Department of Psychology, Faculty of Social and Educational Sciences, Norwegian University of Science and Technology, Trondheim, 7491 Trondheim, Norway; 4Department of Education and Lifelong Learning, Faculty of Social and Educational Sciences, Norwegian University of Science and Technology, 7491 Trondheim, Norway

**Keywords:** sport, affect, resilience, athlete burnout, performance

## Abstract

The current study investigated associations between cognitive components such as psychological resilience and perceived stress, and affective components such as positive and negative affect, and athlete burnout and perceived performance among 670 Norwegian junior athletes attending high schools specialized for sports. A hypothesized model of the relations between the constructs was analyzed by structural equation modeling (SEM). The results in the current study show that athlete resilience is a key in understanding athlete burnout and perceived performance, and that cognitive (perceived stress) and affective reactions (negative and positive affect) are important mediators in this process.

## 1. Introduction

Junior athletes need to cope with multiple stressors on their pathways towards elite sports [1,2,3]. First of all, they are exposed to high physiological loads caused by training and competitions [4]. Secondly, they normally experience social hassles, demanding and high amounts of school tasks, and potential difficulties relating to their peer-groups [5]. Thirdly, they need to participate in competitions and handle competitive stressors [2]. Ultimately, performance enhancements are normally the central concern for ambitious athletes, as athletes themselves and by others are continually evaluating their accomplishments. Thus, junior athletes who aim to become future elite athletes need to learn to cope with such stressors on their pathways towards elite sports [2,3]. Nevertheless, little is known with regard to how psychological factors are associated with the likelihood for athlete burnout or, on the other hand, high perceived performance, among junior athletes.

Stress is a reaction with emotional (biological), physical, cognitive, and behavioral manifestations, that occurs as a result of situations where individuals judge their coping resources to be insufficient [6]. A stressor is a trigger that releases a stress reaction, e.g., during an important competition for an athlete [7]. Stressors in sport can be categorized into competitive stressors [8] and organizational stressors [9]. Competitive stressors are defined as an “ongoing transaction between an individual and the environmental demands associated primarily and directly with athletes’ performance” [2]. Thus, it is how athletes relate to the competitive situation, the appraisals they make, and how they are going to cope with it that defines the transaction. To be competitive, athletes need to prepare mentally, physically, and technically [10]. The situation will be perceived as a significant stressor if they are not well prepared. Occurrences of injuries and illnesses are also significant stressors for athletes, since such occurrences normally make them less competitive [10]. Athletes also need to cope with both internal and external pressures and expectations, and potential rivalry with team members or other competitors in the competitive setting [11]. Importantly, competitions are the key element in elite sports, and athletes with ambitions must learn to cope with such competitive stressors to excel in their sports.

Organizational stressors are defined as “an ongoing transaction between an individual and the environmental demands associated primarily and directly with the organization within which he or she is operating” [2]. Dysfunctional relationships between coaches and athletes, between athletes, and between team members are found to be significant organizational stressors for athletes [3]. Research shows that the working alliance between coaches and athletes can explain both their performances and occurrences of athlete burnout [12,13]. Lack of financial support, personal issues such as lack of necessary nutrition, and team atmosphere are other typical significant organizational stressors [10]. Research indicates that athletes experience more stress that is primarily and directly associated with organizational stressors rather than competitive stressors [8,14]. This is a big paradox, since sport organizations are meant to support athletes in their efforts to develop in their sports.

The theoretical approach that is discussed in the current study shares important similarities with the cognitive activation theory of stress (CATS). CATS focuses on the cognitive element of stress, whereas athletes’ reactions to stressors depend on the cognitive evaluations of the situation and their appraisals of what they can do about it [6]. The stress reaction is defined as positive (eustress) if athletes believe they have the resources to cope with the situation [15,16]. However, stress is defined as negative (distress) when the situation is considered to be novel (e.g., when athletes are favorites in an important competition for the first time), when there is homeostatic imbalance (e.g., when athletes are exhausted because of too much training), and/or when athletes perceive threats (e.g., if they are not well prepared to a competition because of an injury and risk being beaten by other competitors) [17]. Thus, negative stress (distress) in sports results from situational demands that athletes cannot control because of perceived inadequate coping resources.

According to CATS, a negative stressor releases an “alarm” in the body that involves the hypothalamic–pituitary–adrenocortical axis (HPA) [17,18]. The HPA is an important part of the neuroendocrine system that is responsible for releasing stress hormones in the body through interactions between the hypothalamus, the pituitary gland, and the adrenal gland [19]. Ultimately, the body responds differently to eustress and distress [20]. Eustress is associated with positive affect responses, whereas distress is associated with negative affect responses [16]. Eustress stimulates positive affect states such as feeling enthusiastic, active, and alert, in a state that refers to high energy, full concentration, and pleasurable engagement [21,22]. Distress stimulates negative affect responses such as feeling sad and lethargic, in a state that refers to anger, contempt, disgust, guilt, fear, and nervousness [22].

Since the stress reaction is dependent on the appraisal the individual makes of the situation and the available coping resources, the approach is defined as a “transactional” perspective [23]. Thus, stress is an ongoing process that involves individuals transacting with their environments, making appraisals of the situations they find themselves in, and endeavoring to cope with any issues that may arise [23]. Research claims that there are distinct associations between different stress reactions and affect [15,24]. Importantly for sport psychology, these different affect states represent different loads for the individual: Long-term exposure to distress is positively associated with athlete burnout, while eustress is negatively associated with athlete burnout [21,25,26,27]. Importantly, if athletes experience negative stress that lasts over time, the stress might become chronic. Chronic stress normally leads to exhaustion, and athletes might ultimately lose motivation for their sports [28]. This dysfunctional state is defined as athlete burnout [29,30]. Importantly for sport psychology, research claims that the occurrences of athlete burnout are rising in junior sports [31,32]. Therefore, burnout among junior athletes is a big challenge, and it is important to focus on protective factors that might buffer against the potential negative effects from stress in junior sports.

Recent research has identified protective factors that are found to buffer against the potential negative effects from stress in sports [33]. Research shows that athletes who succeed on the stressful pathway towards elite sports have developed a psychological protective mindset that makes them capable of avoiding potential negative consequences from stress, such as athlete burnout and performance impairments [33]. This psychological protective mindset is defined as psychological resilience [33]. Thus, resilience is defined as psychological protective factors or processes that buffer effects of adversity (stress) [34].

Research both from sport [35] and life in general [36,37,38,39,40] points out the existence of key features that protect individuals against negative effects from stress, rather than the absence of such stressors. In general, these protective factors are either within themselves or in their environment. Thus, the protective factors are classified as internal protective factors, such as psychological attributes and family support and cohesion, and external protective factors, such as social support. Psychological attributes include positive personality, optimistic plans for the future, and perception of own social competence [35,40,41,42], whereas family support and cohesion occur when individuals experience a coherence between significant family members in important questions and if the family is supportive in general [37]. External protective factors such as social support refer to the importance of having functional social support systems in athletes’ environments. Individuals who do not have functional social support systems are more vulnerable to stress [43].

The aim of the current study was to investigate possible associations between cognitive components such as psychological resilience and perceived stress and affective components such as positive and negative affect, and athlete burnout and perceived performance, among high-level Norwegian junior athletes attending high schools specialized for sports. Psychological resilience was expected to be positively associated with positive affect and negatively associated with negative affect and perceived stress. Psychological resilience and positive affect were expected to be negatively associated with athlete burnout and positively associated with perceived performance, whereas negative affect and perceived stress were expected to be positively associated with athlete burnout and negatively associated with perceived performance.

## 2. Materials and Methods

One thousand nine hundred and seventeen junior athletes from 27 different Norwegian high schools for elite sports were invited by the authors to voluntarily participate in the current study. Six hundred and seventy out of them completed the data collection. Training is included both on the school schedule and after school every day of the week. The athletes were asked to participate in an online study through an email invitation. The athletes were asked to fill out various self-report instruments measuring psychological variables, such as psychological resilience, affect, perceived stress, athlete burnout, and perceived performance. In addition, they also responded to questions covering demographic variables, such as gender, age, ambitions, type of school, and their sport. Data from the current study are part of a bigger data set that is used in different theoretical approaches. Prior to the data collection, the study was approved by the Norwegian Centre for Research Data.

The variables examined in this study include items and inventories such as age, gender, type of school, type of sport, ambition, and performance level. The psychological measurements that were used in the current study are based on previously developed scales proven to hold both satisfactory validity and reliability. All measurements were used in Norwegian. They are described below in more detail.

**The Resilience scale for adults (RSA).** The RSA is a self-report instrument for evaluating six protective dimensions of resilience in adults: (1) perception of the self, (2) planned future, (3) social competence, (4) family cohesion, (5) social resources, and (6) structured style [37,44,45]. The RSA has 33 items, and examples of items covering the six dimensions are, respectively: “*When something unexpected happens, I often feel perplexed.”, “My plans for the future are difficult to complete, “I feel comfortable together with other people.”, “In my family there is common understanding about what is important in life.”, “I cannot discuss personal problems with anyone.”,* and *“I’m at my best when I have goals that I’m trying to achieve.”* The measurement is tested across different cultures and countries with very acceptable reliability and validity scores [34]. Each item is given a response that ranges from one to seven, where higher scores reflect higher levels of protective factors of resilience. Cronbach’s alphas were 0.79, 0.77, 0.76, 0.79, 0.81, and 0.63 for perception of self, planned future, social competence, family cohesion, social resources, and structured style, respectively.

**The Positive and Negative Affect Schedule (PANAS).** PANAS was used to measure affect based on of two subscales that measure positive and negative affect, respectively [46]. The athletes were asked to look back on experiences they had during the last week and consider to what extent they experienced the different emotions that the different items represent. The items represent ten different emotions used for positive affect (i.e., *excited, strong*, *proud*) and negative affect (i.e., *upset*, *nervous*, *irritable*), respectively, and each item was considered on a 5-point Likert scale from 1 (“*not at all”*) to 5 (*very much*). Earlier studies have reported strong validity [46] for PANAS, and previous research on young athletes has supported the factor structure of PANAS [47]. Cronbach’s alphas were 0.87 and 0.86 for positive affect (PA) and negative affect (NA), respectively.

**The Perceived Stress Scale (PSS-14).** The Perceived Stress Scale (PSS-14) was used to measure perceived stress [48,49]. The PSS is a 14-item scale where the athletes were asked about their thoughts and feelings in general during the last month. Two examples of items are: *“During the past month, how often have you felt that you were unable to control the important things in your life?”,* and *“In the last month, how often have you been upset because of something that happened unexpectedly?”* Thus, the questions measured conditions that were central to the stress experience, such as the degree to which the athletes experienced that their lives were unpredictable, uncontrollable, and overloaded [48,50]. The items were rated on a five-point Likert-type scale from zero (*never*) to four (*very often*). The validity construct is reported to be good for the PSS-14 [48,50]. The reliability for the measurement was 0.84.

**The Athlete Burnout Questionnaire (ABQ)**. The ABQ [26,30] was used to measure athlete burnout. The ABQ comprises three subscales: (1) devaluation of sports participation, (2) a reduced sense of accomplishment, and (3) emotional and physical exhaustion. Each subscale consists of five items that cover each of these dimensions: “*I have negative feelings toward sports*”, “*It seems that no matter what I do, I don’t perform as well as I should*”, and “*I feel so tired from my training that I have trouble finding energy to do other things*”. The athletes were asked to consider to what extent each item reflects their feelings towards their sport participation on a five-point Likert scale ranging from 1 (“*Almost Never*”) to 5 (“*Almost Always*”). A global burnout score was computed as described by Raedeke and Smith [51]. The reliability and the factorial and convergent/divergent validity of the ABQ are supported in previous research [30,31,52]. Cronbach’s alphas were 0.85, 0.86, and 0.76 for emotional and physical exhaustion, devaluation of sports participation, and reduced sense of accomplishment, respectively.

**Perceived performance in sport.** The athlete satisfaction questionnaire was used to measure athletes’ perceived satisfaction with their own progress in sport [53]. The scale includes 4 items that cover the athletes’ perception of their absolute performance (“I am satisfied with the degree to which I have reached my performance goals during the season”), improvements in performance (”I am satisfied with the degree of development of my skill level”), and goal achievement (“I am satisfied with my goal achievements the last period”). The items were considered on a 7-point scale ranging from 1 (not at all satisfied) to 7 (extremely satisfied) based on their experiences from training and competitions in the current period. Cronbach’s alpha was 0.92 for the measurement.

### Data Analysis

Firstly, data were analyzed by examining the correlations between the variables using the Pearson correlational coefficient. Then, descriptive statistics such as statistical means, standard deviations, and maximum and minimum values and the Cronbach’s alpha were computed. These analyses were conducted with the SPSS 25 program (IBM Corp., New York, United States).

Thirdly, the data were analyzed by structural equation modeling (SEM) using the AMOS 25 program (IBM Corp., New York, United States) [54]. SEM uses a confirmatory approach to the analysis, where a hypothesized model of the relations between the theoretical constructs is tested statistically to determine the extent to which it is consistent with the data [55]. The result is referred to as the goodness of fit of the theoretical model. If the goodness of fit is adequate, the plausibility of the proposed relations among the constructs is supported.

## 3. Results

From the 1917 participants, 670 (49.3% males and 51.7% females) completed the data collection, which gives a response rate of 35.5%. The sample had a mean age of 18 years (ranging from 17 to 20 years) and practiced a variety of sports, with football (18%), handball (18%), cross country skiing (11%), biathlon (9%), ice-hockey (5%), alpine skiing (5%), cycling (5%), and track and field (4%) being those most frequently reported. Seventy eight percent of the junior athletes in the current study had ambitions to become future elite athletes in their sports, whereas 22% did not [56].

Table 1 shows the correlations between the study variables as well as the possible maximum scores, statistical means, standard deviations, and Cronbach’s alphas.

One approach that has been used to interpret burnout inventory scores is to develop norms by dividing the distribution into thirds [57]. Scores among college students in the upper-third reflect high degrees of burnout (cutoff scores for exhaustion, devaluation, and accomplishment are 3.00, 2.40, and 2.60, respectively), scores in the middle-third reflect average tendencies of burnout (cutoff for exhaustion, devaluation, and accomplishment are 2.20, 1.60, and 2.00, respectively), and those in the bottom-third reflect low burnout scores [26]. Table 1 shows that the mean values reflect average burnout scores in the upper level on exhaustion and devaluation, whereas the mean values for accomplishment scores are reflecting high burnout. Frequencies statistics show that 162 (101 males and 61 females) junior athletes reported exhaustion scores in the upper-third (24%), 261 (182 males and 79 females) reported devaluation scores in the upper-third (39%), and 368 (267 males and 101 females) reported accomplishment scores in the upper-third (55%).

Introductorily, we outlined a complex process model incorporating several factors which could affect burnout and perceived performance among athletes, namely resilience, perceived stress, and positive and negative affective states. The correlation analysis in Table 1 shows that all variables in the current study were weakly or strongly associated in either a positive or a negative direction. In order to gain a better understanding of these complex patterns of associations, we performed path analyses based on the theoretical model presented in the introduction (Figure 1).

There were several indirect effects in the model. Resilience had indirect effects on positive affect (*z* = 0.17, *p* < 0.05), negative affect (*z* = −0.38, *p* < 0.05), perceived performance (*z* = 0.30, *p* < 0.05), and burnout (*z* = −0.33, *p* < 0.05). These results emphasize the endogenous role of resilience on many potential psychological factors among athletes. Furthermore, perceived stress had indirect effects on perceived performance (*z* = −0.23, *p* < 0.05) and burnout (*z* = 0.31, *p* < 0.05). We also controlled for gender in the model, finding that gender had an effect on perceived stress (*ß* = 0.19, *p* < 0.01), but all other paths were nonsignificant. Further, the inclusion of gender in the model had no substantial impact on the paths in the model (all with/without gender *ß*-discrepancies were below 0.02, indicating that the interplay of variables in the model are nongender-specific.

Assessments of model fits were judged according to criteria suggested by Hu and Bentler [58,59]. We used the chi-square test, the comparative fit index (CFI), the Tucker-Lewis Index (TLI) and the root square mean error of approximation (RMSEA) provided in AMOS. In general, CFI and TLI values above 0.95 and RMSEA below 0.08 indicate that the model fits the data adequately. We also tested indirect effects in the model [60]. The overall fully saturated model did of course explain all variance in the model: (χ^2^ (0, *N* = 670) = 0.00, *p* < 0.001. We then trimmed the model by removing nonsignificant paths and covariances starting with the path with the highest *p*-value, using a *p* < 0.01 criterion adjusted to the N in the sample [61]. The modified model had adequate model fits: (χ^2^ (5, *N* = 670) = 15.50, *p* = 0.008, CFI = 0.993, TLI = 0.980, RMSEA = 0.056.

## 4. Discussion

The theoretical model in the current study of associations between cognitive and affective components of stress, athlete burnout and perceived performance, was mainly confirmed. The results in the current study confirm the hypotheses, whereas all relations are significant except for the expected associations between resilience and negative affect and resilience and perceived performance, and perceived stress and athlete burnout and perceived stress and perceived performance. The model fit scores are good. Based on the results in the current study, it can be argued that athlete resilience is a key in understanding the athlete burnout syndrome and perceived performance, and that cognitive and affective responses are mediating effects.

### 4.1. Athlete Resilience: A Key Buffer in Athlete Burnout

First of all, the relatively high percentage of junior athletes who score in the upper third of burnout scores in the current study is alarming and shows the importance of studies that investigate the burnout syndrome (24%, 39%, and 55% for exhaustion, devaluation, and reduced accomplishments, respectively). The results in the current study point out the importance of athlete resilience in understanding athlete burnout and perceived performance. The tested model shows a strong, negative association between resilience and perceived stress. Thus, as theory claims, highly resilient athletes will experience less stress when they experience life- or sport-related adversities, in comparison with less resilient athletes [33,34]. The strong negative association between resilience and perceived stress can be attributed to the psychological reactions that occur when athletes experience stressors. It is how athletes relate to potential stressors that decides their perceived stress experience. For instance, if athletes think they are capable of executing necessary actions to overcome adversity (perception of self), they will not experience stress to the same extent as if they do not think they are capable of doing that. If athletes think they are not able to do what is necessary to achieve their goals, it might be experienced as a self-discrepancy, either a specific discrepancy, e.g., within their sports, or a discrepancy related to their global self [62,63]. If athletic performances are exclusively related to athletes’ identity, they will struggle to keep a positive global self when exposed to difficulties in their sports [64]. Interestingly, the strongest correlations in the current study are the negative correlations between perception of self and perceived stress. This might indicate that keeping a positive self-perception might be crucial in preventing possible negative consequences from stress, such as athlete burnout, and in influencing perceived performance positively.

### 4.2. The Cognitive Moderator

Correlations between the dimensions of resilience and negative affect vary from weak to high. Perception of self, planned future, and social resources are the dimensions with the strongest correlations, whereas social competence, family cohesion, and structure are the variables with weak correlations. However, there is no significant association between resilience and negative affect as the theoretical model in the current study suggested. A possible attribution can be found in earlier research that claims that people react differently when they experience adversities and setbacks, and not everyone gets stressed and exhausted from it [65]. A plausible explanation is therefore that it is how athletes relate to adversities that makes the difference, not the adversity itself. Thus, the relationship between resilience and negative affect is mediated through perceived stress. The cognitive component is therefore essential, and the cognitive appraisal therefore predicts the affective stress response, which is consistent with theory and research [6,17,21]. Importantly, one can assume that athletes with higher levels of resilience will experience less stress, which in turn reduces the exposure to chronic stress, and as well as the likelihood of experiencing burnout.

The current study did not confirm the expected association between resilience and perceived performance. This is an interesting finding that might be attributed to the importance of resilience to withstand adversity and difficulties. Thus, athletes can withstand the experience of stagnation and difficulties in their sports and still be resilient (e.g., not being satisfied with their performances). However, this needs to be studied in longitudinal studies.

### 4.3. The Affective Moderator

Interestingly, the current study shows that it is the affective components that are strongly associated with athlete burnout and perceived performance. Thus, it is how athletes cognitively (perceived stress) relate to difficulties and adversity that predicts their affective reactions, and it is the affective reactions that influence the impact on burnout and perceived performance. The current study did not find any significant associations between perceived stress and athlete burnout, and perceived stress and perceived performance. However, the association between positive affect and athlete burnout is strong and negative, whereas the association between positive affect and perceived performance is strong and positive. The associations between negative affect and athlete burnout are moderate and positive, while the associations between negative affect and perceived performance are weak and negative. When athletes are exposed to adversity or difficulties and expect that the outcome of the situation will turn out well for them, their affective reaction will be positive [20,24]. In such cases, athletes believe they have the necessary resources to cope with the demands of the situation [15]. However, if they do not think they have the necessary coping resources to handle the situation, the affective response will be negative. Importantly, it is the perception of themselves in the situation that stimulates the affective reaction, which underlines the importance of mental resilience in understanding athletes’ perceived stress level and the affective stress response. Thus, emotions seem to play an important role in the work to develop junior athletes towards elite sports.

### 4.4. Limitations

Even though the current study has several interesting results, it also has several limitations. First of all, longitudinal studies are needed to fully investigate both direct and indirect relationships and how these develop over time. Secondly, the data in the current study were based on self-reporting measures and how these self-report instruments accurately reflect the variables is unknown. Future studies should combine self-reported data with data obtained in a more objective manner.

## 5. Conclusions

The current study indicates that athlete burnout and perceived performance can be explained by athlete resilience, and that cognitive and affective reactions are important mediators in this process. Finally, the current study indicates that affect plays an important role in the development of athlete burnout and perceived performance. However, affect is associated with perceived stress and perceived stress is associated with athlete resilience. Thus, interventions that aim to build athlete resilience should be of high importance in junior sports.

## Figures and Tables

**Figure 1 sports-07-00171-f001:**
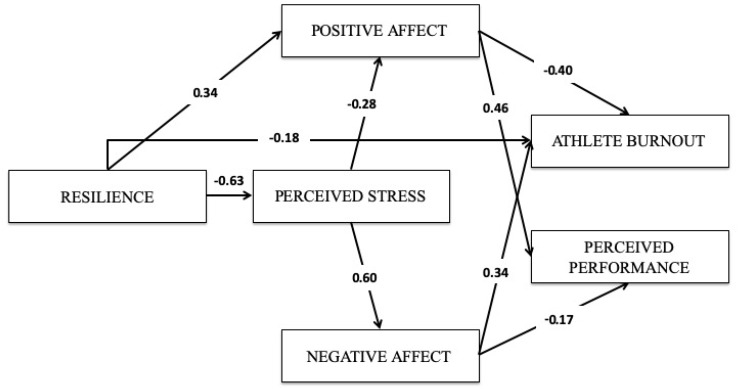
The observed significant relations between the variables in the theoretical model in the study.

**Table 1 sports-07-00171-t001:** Correlations, descriptive statistics, and Cronbach’s alphas of the variables, n = 670.

	1	2	3	4	5	6	7	8	9	10	11	12	13
1. RS-Self	-												
2. RS-Planned Future	0.62 *	-											
3. RS-Social Competence	0.43 *	0.32 *	-										
4. RS-Family Cohesion	0.41 *	0.38 *	0.22 *	-									
5. RS-Social Resources	0.47 *	0.46 *	0.44 *	0.66 *	-								
6. RS-Structure	0.35 *	0.43 *	0.22 *	0.31 *	0.33 *	-							
7. PA-Positive Affect	0.47 *	0.49 *	0.28 *	0.27 *	0.32 *	0.31 *	-						
8. NA-Negative Affect	−0.45 *	−0.36 *	−0.18 *	−0.25 *	−0.35 *	−0.19 *	−0.27 *	-					
9. Perceived Stress	−0.68 *	−0.56 *	−0.30 *	−0.37 *	−0.44 *	−0.33 *	−0.49 *	0.60 *	-				
10. ABQ-Exhaustion	−0.44 *	−0.43 *	−0.21 *	−0.28 *	−0.34 *	−0.34 *	−0.42 *	0.51 *	0.53 *	-			
11. ABQ-Devaluation	−0.28 *	−0.39 *	−0.17 *	−0.25 *	−0.29 *	−0.32 *	−0.52 *	0.37 *	0.37 *	0.63 *	-		
12. ABQ-Accomplishment	−0.46 *	−0.43 *	−0.25 *	−0.24 *	−0.32 *	−0.22 *	−0.52 *	0.43 *	0.49 *	0.50 *	0.51 *	-	
13. Perceived performance	0.34 *	0.38 *	0.16 *	0.16 *	0.25 *	0.24 *	0.51 *	−0.29 *	−0.35 *	−0.31 *	−0.31 *	−0.61 *	-
Mean	4.91	5.20	5.12	5.76	6.13	5.12	36.3	22.8	24.5	2.35	2.20	2.67	4.71
Standard deviation	1.16	1.27	1.10	0.99	0.86	1.20	6.63	7.18	7.70	0.81	0.81	0.81	1.15
Maximum	7	7	7	7	7	7	50	50	56	5	5	5	7
Minimum	1	1	1	1	1	1	12	10	0	1	1	1	1
Cronbach‘s alpha	0.79	0.77	0.76	0.79	0.81	0.63	0.87	0.86	0.84	0.85	0.86	0.76	0.92

*Note.* * *p* < 0.01.

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
