# Peer review of "Burnout and Perceived Performance Among Junior Athletes—Associations with Affective and Cognitive Components of Stress"

_sports, 2019, doi:10.3390/sports7070171_

Round 1
Reviewer 1 Report
This manuscript submitted to Sports presents an interesting study investigating a theoretical framework between affective and cognitive components and athlete burnout and perceived performace. 670 participants were included in the study. I think the study has some minor issues I state point-by-point:
- Row 211 -220: I think this should be included in the results section, however model fits are not adequatly displayed there.
- Row 245 insert space between "Table 1" and "shows"
- Table 1 correct Cronbach's alpha NOT Chronbach's as in the last table row
- What is an explanation why there is no siginifcant relation between reslilience and negative affect; but between resilience and positive affect. This should be discussed; authors argue that this is moderated by perceived stress, but discussion should parse further.
- Authors did control for gender, but did not discuss the influence of gender in the discussion.
In my opinon the manuscript lacks a limitation section, as this study defenitely has strenghts and weaknesses.
Author Response
Thank you for such positive feedback on our work.
The information about the model fit results are now moved to the results section as suggested (after the figure).
The space is inserted between table1 and shows.
Cronbach's alpha is corrected.
The explanation why there is no significant association between resilience and negative affect is revised.
Tar authors do not think it is relevant to discuss gender in the discussion section since it was not the main purpose of this study.
The limitation section is put into the discussion section.
Reviewer 2 Report
Dear authors,
I want to congratulate to authors for this study. In the study authors have analysed the effect of burnout and perceived performance with affective and cognitive components of stress in a sample very important according to my opinion. The size of the sample is very high and the proportion of different sports modalities beside an interesting and appropriate test make the conclusion of this study very important in Sport Psychology.
I think this paper is very well, but authors must correct the next considerations:
- Lines 31-37: include, at least, a reference.
- Lines 127: Delete this figure. A introduction is a localization don´t adequate for a figure.
- Lines 264-266: I think is not necessary to repeat the objective of the study in this sections. It would be better staring with the mail results of the study “The theoretical…”
- Authors must borrow the titles of the discussion (lines 281, 303 and 322)
As opposed to the main text, the abstract of the paper is wrong and it must be corrected. First of all, it´s necessary to include the correct aspect s of the sample. Also, it´s necessary to include the tests employed and the main results (including the main results, including valued and p-values) and a summary of the conclusions section.
Author Response
The authors would like to thank you for such constructive feedback on our work.
Two references are included in the section.
The figure is deleted.
The introduction to the discussion section is revised according to the suggestion.
The titles of the discussion is borrowed.
The abstract is revised.